# How Implicit Attitudes toward Vaccination Affect Vaccine Hesitancy and Behaviour: Developing and Validating the V-IRAP

**DOI:** 10.3390/ijerph19074205

**Published:** 2022-04-01

**Authors:** Luca Simione, Monia Vagni, Tiziana Maiorano, Valeria Giostra, Daniela Pajardi

**Affiliations:** 1Institute of Cognitive Sciences and Technologies, CNR, 00185 Rome, Italy; 2Department of Humanities, University of Urbino, 61029 Urbino, Italy; monia.vagni@uniurb.it (M.V.); t.maiorano@campus.uniurb.it (T.M.); valeria.giostra@uniurb.it (V.G.); daniela.pajardi@uniurb.it (D.P.)

**Keywords:** vaccine hesitancy, vaccine propensity, implicit attitudes, COVID-19, IRAP

## Abstract

Vaccination is one of the most important ways of fighting infectious diseases, such as COVID-19. However, vaccine hesitancy and refusal can reduce adherence to vaccination campaigns, and therefore undermine their effectiveness. Although the scientific community has made great efforts to understand the psychological causes of vaccine hesitancy, studies on vaccine intention have usually relied on traditional detection techniques, such as questionnaires. Probing these constructs explicitly could be problematic due to defense mechanisms or social desirability. Thus, a measure capable of detecting implicit attitudes towards vaccination is needed. To achieve this aim, we designed and validated a new test called the Vaccine-IRAP, or V-IRAP, which is a modified version of the original Implicit Relational Assessment Procedure, or IRAP, task. The V-IRAP allows the unspoken reasons behind vaccine hesitancy to be investigated, and is able to distinguish between positive and negative beliefs on vaccination. The test was assessed in a sample of 151 participants. The V-IRAP showed good internal reliability and convergent validity, with meaningful correlational patterns with explicit measures. Moreover, it revealed incremental validity over such explicit measures. Lastly, the V-IRAP was able to shed light on the implicit attitudes involved in vaccine refusal, revealing negative attitudes relative to vaccine-related risks in non-vaccinated participants. Overall, these results support V-IRAP as a sensitive and reliable tool that could be used in future studies on implicit attitudes toward vaccination.

## 1. Introduction

Attitudes towards vaccines can be defined as propensity, indicating a position ranging from passive acceptance to active demand, and hesitancy or refusal, indicating, respectively, a position of uncertainty about getting vaccinated or a position against getting vaccinated [1]. The different intermediate positions between the two extremes of this scale not only have a cognitive and rational connotation, but also an emotional and unconscious one [2]. Since the beginning of the COVID-19 pandemic, science has been looking for a vaccine formula to combat COVID-19 [3,4,5], and governments have supported the academic community and the pharmaceutical industry [6,7] because vaccines were considered the main instrument to address virus diffusion [8]. However, vaccination has to be accepted by a large part of the population to be effective. Therefore, local governments and international organizations have planned campaigns to promote vaccination or have even made vaccination mandatory, for example by introducing vaccination certificates or COVID-19 green certification as tools to access social events and public services. As it has valuable social and health consequences, understanding the psychological variables associated with attitudes towards vaccination was one of the main topics of research on the COVID-19 pandemic during 2021.

Vaccine hesitancy is a worldwide phenomenon, and several anti-vax protest movements have gained visibility through the media and social networks [9]. Vaccination hesitancy is considered a multifactorial phenomenon influenced by several factors: cognitive, psychological, socio-demographic, political, and cultural [10]. Opposition to vaccines is an attitude antecedent to the COVID-19 pandemic [1] that has spread especially in the last 10 years [11]. Consequently, in 2019, the WHO declared vaccine hesitancy as one of the ten most serious threats to health [12]. An important milestone in the spread of vaccine refusal and hesitancy was the H1N1 influenza vaccine in 2009 and 2011, due to suspicions around its safety and distrust of public health institutions [8]. 

Many studies have tried to investigate the underlying determinants of vaccine hesitancy during the COVID-19 pandemic. The literature shows that younger age, low education level, and low income were common sociodemographic factors associated with increased vaccine hesitancy [13,14,15]. Regards demographic variables, also female sex was associated with increased hesitancy for COVID-19 vaccines [16]. Perception risk associated with COVID-19 infection was also related to vaccine hesitancy, as lower perception of danger leads to higher belief in conspiracy theories, and thus to lower vaccination acceptance [13,15]. In another rapid systematic review and meta-analysis a similar pattern of results emerged, with male sex and higher perceived risk of COVID-19 infection as predictors of increased acceptance of the vaccine [17]. Regarding psychological variables and beliefs, a review of 82 studies [14] reported that the most common factors affecting vaccination hesitancy include perceived vaccine efficacy and side effects, mistrust in the healthcare system, religious beliefs, and trust in information sources. Misinformation, the use of social media or the Internet as a main source of information, and the lack of widely accessible information on vaccination were also associated with vaccine hesitancy [15]. 

Along with these more generic factors, there are also determinants of vaccine hesitancy that were specific of the COVID-19 scenario, such as distrust in the rapid development of vaccines, considering the vaccine more dangerous than COVID-19 or considering COVID-19 as harmless, believing oneself to be already immunized, and doubt about the provenience of vaccine [18]. In particular, the development of COVID-19 vaccines was very rapid, so people felt that there was a higher risk of this vaccination in comparison to previous ones, but at the same time knew less about its benefits and costs [19]. The literature on COVID-19 underlined a direct relationship between the perception of the risk of being infected and the propensity to get vaccinated [20,21,22,23]. This factor can cause the level of hesitancy to rise six-fold in people who are confident that they will not be infected [24]. Consistent with this, Dror et al. [3] found that being a physician working on the front line with COVID-19 patients led to a greater propensity to get vaccinated than the general population, in the light of the experience of the severity of the epidemic and as a way of keeping safe. Meanwhile, healthcare personnel who were not directly in contact with these patients reported greater hesitancy. However, a generally higher acceptance of vaccination among healthcare workers with respect to the general population was reported in the literature [25].

Perception of risk depends on individual attitudes. According to the socio-cognitive model proposed by Eberhardt and Ling [26], people are more likely to engage in protective behavior when they believe that not acting poses a risk to themselves and that protective behavior reduces the threat. On the contrary, people reporting vaccine hesitancy think that it does not make sense to prevent an infection perceived as not harmful by using a vaccine that was, on the contrary, considered dangerous [27]. Thus, a greater tendency to prioritize the risks of the disease relative to the risks of side effects was associated with a higher likelihood of intention to vaccinate [28]. As risk perception was strongly affected by emotional factors, Hornsey, Harris, and Fielding [29] underlined how communication that acts only on a cognitive level and tries to persuade only through information is ineffective because of the complex and inexplicable roots of this attitude. 

Trust and distrust in the vaccine were common manifestations in all countries around the world, and involved the entire population. The data reported by Stolle et al. [30] described a situation in which 80% of the world’s population is in favor of the vaccine, with less than 10% of subjects being anti-vax, highlighting a margin of 10% of undecided subjects in their attitude. A recent meta-analysis [11] reported a medium-high acceptance level in Italy. However, the results of the different studies were affected by both the evolution of the pandemic scenario and the characteristics of the specific sample. 

In Italy, the vaccine acceptance rate was 77.3% in April, 70.8% in June, and it reached 53.7% in September 2021 [13], but at the beginning of 2022, 90% of the population had received at least one dose of the vaccine (Italian National Institute of Health, 14 January 2022) [31]. These data suggest that the population was getting vaccinated, even if it could not highlight their motivations and deeper attitudes. Recent studies showed how motivations related to existential anxiety and inner beliefs were involved in vaccine hesitancy [2,32]. Probing these constructs explicitly, for example, with self-report measures, could be problematic due to defense mechanisms associated with them [33]. Thus, other measures capable of detecting these implicit attitudes are needed to conduct more compelling research on vaccine hesitancy. 

In the scientific literature reported so far, a heterogeneity in vaccine hesitancy assessment emerged, pointing out the need for standardization in its assessment [15]. Furthermore, in these studies, people’s opinions about COVID-19 and its related vaccine have mainly been investigated through more traditional detection techniques, such as questionnaires with explicit questions against which people enact rational forms of control over their process of answering. In these cases, the response of the participants depends on how the question is formulated, on its content, and the modality of the response: dichotomous (yes/no) or on a Likert scale (from agree to disagree). In the response process, however, other psychological factors or causes that influence the response may intervene, such as conformity, complacency, fear of the judgment from others, oppositionality, etc. Therefore, what has been declared does not always fully correspond to the beliefs and real attitudes of the people. In other words, through an implicit test, it is also possible to detect the unspoken.

Attitude has an implicit dimension that is of fundamental importance in understanding the transition to a certain behavior or another and has been extensively studied in the literature in various contexts as an explanation of stereotypes. Greenwald and Banaji [34] defined implicit attitudes as “introspectively unidentified (or inaccurately identified) traces of past experiences that mediate favorable or unfavorable feelings, thoughts, or actions toward social objects” (p. 8). The literature has widely demonstrated the limits of questionnaires in detecting such attitudes, as they are at risk of deception and self-presentation strategies [35]. For this reason, the Implicit Association Test (IAT) [36] has been very successful and has been applied to detect stereotypes and prejudices in different contexts, above all regarding racial stereotypes. One of the main limitations of IAT and its variants is that they focus on grasping the association between stimuli or events, but do not assess the nature or direction of such, as well as not assessing a complex structure of associations [35]. To overcome this limitation, an alternative test was developed, which is the Implicit Relational Assessment Procedure, or IRAP [37]. The IRAP was based on the Relational Frame Theory (RFT) and aimed to assess the strength or probability of individuals’ relational responses. The participants were then asked to associate verbal stimuli in an automatic and non-deliberative way, which therefore drawn on an implicit register. The model is based on linguistic stimuli and aims to highlight the relationship between targets and stimuli while varying relational rules. A typical example is the self-esteem IRAP [38], in which the labels ‘self’ and ‘others’ should be associated with positive or negative targets. Two association rules determining how labels and targets should be associated were alternated in a series of blocks of trials. The test quantifies the implicit attitude toward one rule or the other as the time required for participants to respond accordingly with the rules. For example, if the association of ‘self’ with positive targets is faster than the association with negative targets, an implicit bias towards a self-positive relational representation could be inferred. 

On this basis, the present study sought to detect the implicit associations that people have with COVID-19 and the vaccine by using a modified IRAP test. The implicit associations were then related to the explicit declaration of propensity or refusal to the vaccine. To achieve this goal, we adapted a tool, the IRAP, to the theme of the COVID-19 vaccine. Then, we developed and validated a new test called the Vaccine-IRAP or V-IRAP. Within this framework, we were able to estimate the implicit attitude towards the positive aspects (safety and usefulness) and negative aspects (danger and risk) of vaccination. We also related these implicit measures to an explicitly evaluated vaccination hesitancy/acceptance and vaccination behavior, that is to the actual adherence to the vaccination campaign. The test would also allow the understanding of which component between considering the vaccine as not useful (low vaccine-positive association) or as harmful (higher vaccine-negative association) was prominent in determining vaccination hesitancy/refusal. From the reviewed literature, we hypothesized in particular that non-vaccinated participants would report both a belief that vaccination was not safe [18]—that is, they would report a bias towards the association between the vaccine and negative aspects—and a belief that vaccination is not useful [14]—that is, they would report a reduced bias towards the association between the vaccine and positive aspects. Moreover, we hypothesized that implicit attitudes were predictive of vaccination hesitancy and that they also increment the capability of predicting vaccination behavior with respect to the mere explicit measure. 

## 2. Materials and Methods

### 2.1. Participants

For this study, we enrolled 151 volunteer participants through snowball sampling on social media during May, June, and July 2021. Regarding demographic variables, all participants were Italian, 123 participants were female, and 28 were male, and the mean age was 38.87 years (SD = 13.57). Overall, 77% of our participants were in a stable relationship, 3% in an unstable or conflictual relationship, and 20% were single. A total of 66% reported having no children, while 14% reported having one child, 17% had two children, and 3% had three or more children. Regarding their working conditions, 47% were autonomous workers, 31% were employees, 18% were students, and 4% were unemployed. In all, 56% reported working as usual, 23% to be engaged in remote work, while 21% did not work (9% of which had lost their job due to the pandemic). Regarding their medical and psychological conditions, 84% reported no medical condition, while 16% reported one (15%) or two (1%) pathologies linked to worst COVID-19 outcomes. Only four participants reported psychological conditions, such as anxiety, depression, OCD, and schizoid disorder. Regarding vaccination for COVID-19, 118 (78%) of the participants had had at least one dose of the vaccine, while 33 (22%) had not. Among the vaccinated participants, 76 had received Comirnaty (Pfizer-BioNTech), 40 Vaxzevria (Astrazeneca), one the Moderna vaccine, and one Janssen (Johnson & Johnson). All participants reported normal or corrected to normal vision.

This study was part of a larger project in which participants compiled a battery of questionnaires and then performed the IRAP task. As the task was optional, in this article we reported the results relative only to the participants who completed the IRAP task.

### 2.2. Procedure

The entire procedure was administered online through PsyToolkit, a reliable and widely used web-based platform for collecting experimental data from the questionnaire and the reaction time [39,40]. The materials were administered through a series of successive forms. First, participants read and signed the informed consent to participate. Secondly, they completed a form about socio-demographic information. Thirdly, we presented a battery of questionnaires investigating psychological symptoms and distress. At the end of the questionnaires, participants had to answer some questions about their vaccination status. Afterwards, they could go on to the IRAP task or exit the procedure. If they continued the task, the experimental procedure was presented starting from general instructions, followed by practice blocks, and then by experimental blocks (see Materials section for details). As reported, 151 participants completed both of the questionnaires and the IRAP task, while 224 completed only the questionnaires (their data are not reported here). At the end of the procedure, participants were briefly debriefed about the entire experiment and thanked. 

All data were collected in a completely anonymous format. Informed consent was obtained from all individual participants included in the study. Ethical approval for this study was granted by the Ethics Committee of the University of Urbino and all procedures performed were under the ethical standards of the 1964 Helsinki Declaration. 

### 2.3. Materials

#### 2.3.1. The Online Survey

The survey included a variety of questionnaires for psychological symptoms and distress that are not reported here. We also asked for sociodemographic information, such as sex, age, education level in years, present level, nationality, relationship status, number of children, religious belief, working condition, and presence of psychological or medical conditions. In particular, medical problems associated with an increased risk in the event of COVID-19 infection were investigated. We computed a health index as the sum of the following eight possible conditions measured using a checklist: cardiovascular diseases, diabetes mellitus, hypertension, chronic pneumopathies, neoplasms, immunodeficiencies, hematological pathologies, and neuromuscular diseases.

After completing the questionnaires, participants reported whether they were vaccinated against COVID-19, the type of vaccine (among Comirnaty, Moderna, Vaxzevria, Janssen, Sputnik V), and their degree of agreement (from 1, ‘completely disagree’ to 5, “completely agree”) with the following statement ‘I think the vaccine I had is safe and I would do it again’. Instead, participants who did not have a COVID-19 vaccine reported which vaccine they would prefer (among Comirnaty, Moderna, Vaxzevria, Janssen, Sputnik V), and their degree of agreement (from 1, ‘completely disagree’, to 5, ‘completely agree’) with the following statement: ‘I think vaccination is safe and I would like to do it as soon as possible’.

#### 2.3.2. The Implicit Relational Assessment Procedure (IRAP) Task

The IRAP is a computerized task in which participants have to respond to a combination of two verbal stimuli indicating if their association is true or false, based on a set of pairing rules. The stimuli included two label stimuli and two lists of targets. In our study, the two labels were ‘COVID-19’ and ‘VACCINE’, while the targets were divided into two 5-item lists, one including adjectives indicating a positive attitude or trust, e.g., good, healthy, useful, harmless, and reassuring (in the original Italian version: buono, salutare, utile, innocuo, rassicurante), and the other including negatively connotated adjectives, e.g., bad, deadly, harmful, dangerous, and frightening (in the original Italian version: cattivo, mortale, dannoso, pericoloso, spaventoso). The targets were selected through an iterative procedure. First, a large set of possible targets were proposed by collecting contributions from the research team. Then, the whole pool of candidate targets was evaluated by the research team again and a restricted list of ten targets per list was formulated. These lists were then evaluated by two experts who were not part of the research team. Based on their suggestions and evaluations, a final set of targets was defined as described. While the association between COVID-19 and terms such as good or healthy could be considered bizarre, please consider that the task aimed to properly reveal anti-intuitive or unusual associations. Moreover, as we were mainly interested in the responses associated with the labels ‘VACCINE’ (see Results section), such associations with COVID-19 were not problematic to us.

In each IRAP trial (Figure 1), four stimuli were presented simultaneously: one of the labels was presented at the top of the screen, one of the targets in the center, and the two response options (‘true’ and ‘false’) were presented at the bottom of the screen, one on the left and one on the right. The left response option was associated with the key ‘D’, while the right response option was associated with the key ‘K’. The left–right positions of the response options changed randomly on a trial-by-trial basis. A correct response, i.e., a response in accordance with the association rule given at the beginning of the block, cleared the screen for 400 ms and then another trial was presented. In the case of a wrong response, i.e., a response not in accordance with the association rule given at the beginning of the block, a red X was presented between the two response options until the participant gave the correct response. Following the label–target association given, IRAP included four trial types (see Figure 1): COVID-negative, COVID-positive, VACCINE-negative, and VACCINE-positive.

Each block of trials included 20 trials, i.e., one for each label–target pair. To associate labels and targets, an association rule was provided at the beginning of each block of trials. Two rules existed, one called ‘consistent’ or rule A, in which participants had to answer as if the vaccine were positive and COVID-19 negative, and the other called ‘inconsistent’ or rule B, in which participants had to answer as if the vaccine were negative and the COVID-19 positive. For each block, the consistent or inconsistent rule was applied. The two rules were alternated between pairs of consecutive blocks, so that the first block was a consistent block (rule A) and the second was an inconsistent block (rule B). The order in which this sequence was presented (consistent followed by inconsistent or inconsistent followed by consistent) was counterbalanced between the participants. 

To familiarize participants with the IRAP, the task started with a practice phase, including two to six practice blocks. To enter the experimental phase, participants had to reach a standard of 80% correct responses and a median RT of less than 2000 ms in both a consistent and an inconsistent block. After each block of trials, participants were informed of their performance and if they had failed to meet either the accuracy or RT criterion or both. If the criteria were not met, the same block was presented again. Participants who did not meet the criteria after a maximum of six blocks (three consistent and three inconsistent) were thanked and their experimental session was concluded. Participants who met the criteria for both consistent and inconsistent practice blocks proceeded to six test blocks. These blocks were similar to the practice ones, except that only performance information was presented at the end of each block, without any criteria-related information. After completing the six test blocks, the experiment ended with a message of thanks. 

### 2.4. Data Analysis

We analyzed IRAP data with the D-score algorithm described in Barnes-Holmes et al. [37] and Hussey et al. [38]. Analyses were performed only on the data from the six test blocks. To obtain a final dataset including only participants who adequately performed the task, we applied speed and accuracy constraints as indicated in Barnes-Holmes et al. [37]. Due to the online procedure, we relaxed such constraints as follows. For participants who displayed under 70% accuracy overall, the six blocks were discarded. We also discarded participants with a median RT longer than 2500 ms. Lastly, we removed participants who answered faster than 300 ms in at least 10% of trials. D-scores were computed by subtracting mean RTs for different types of trials in block pairs and dividing this difference by the total standard deviation for the same block pair. After computing four D-scores for each block pair, an overall set of four D-scores were computed by averaging the three sets obtained for the three pairs of blocks. In our study, positive D-scores indicated a bias toward the consistent rule, i.e., vaccine is good/COVID-19 is bad, while negative D-scores indicated a bias toward the inconsistent rule, i.e., vaccine is bad/COVID-19 is good.

D-scores were analyzed as in a typical IRAP-task [37,38]. The internal consistency of the V-IRAP was assessed by calculating a split-half reliability score. In particular, a Spearman–Brown split-half correlation analysis was conducted between the D-scores obtained in even and odd trials. Then, an ANCOVA was implied to test the main effects and interaction between trial types while controlling for covariates such as age, sex, and education level. The ANCOVA was followed by Holm–Bonferroni post hoc tests. To test the significance and the strength of each bias, the four IRAP D-scores were compared to zero with one-sample *t*-tests. Each analysis reported the average score, 95% confidence intervals, statistics of the *t*-test, and Cohen’s effect size as d. 

The same analysis pattern was then replicated separately for participants who had received or had not received COVID-19 vaccination. The D-scores obtained by the two groups were compared through an ANCOVA controlling for age, sex, and education level, followed by Holm–Bonferroni corrected post hoc tests. A set of one-sample *t*-tests were also conducted for each group separately.

Subsequently, an incremental validity analysis was performed to assess the effective contribution of the implicit measures not obtained by the explicit measure. A hierarchical regression model was conducted on vaccination behavior (0 = not vaccinated; 1 = vaccinated). In the first step, the covariates were included in the model (age, sex, and education level); in the second step, the explicit measure of vaccine acceptance was also included: in the third step, the two D-scores obtained in the VACCINE-positive and VACCINE-negative IRAP trials were finally included. For each model step, both the unstandardized and standardized coefficients of each regressor were reported along with the fit statistics as R^2^. Each coefficient or statistic was reported with its 95% confidence intervals estimated over 1000 bootstrap samples. Furthermore, each step was compared with the previous to assess whether the inclusion of explicit or implicit measures significantly increased the fit of the model. This analysis aimed to show the predictive capacity of both implicit and explicit beliefs about vaccination behavior. Pearson’s correlations between these variables were reported before conducting this regression analysis.

Finally, we tested hypothesized mediational pathways through structural equation modeling. In this model, we tested the effect of implicit biases on vaccination mediated by explicit vaccine propensity, while controlling for demographic variables such as sex, age, and education level. The model analysis was conducted using the Huber–White robust standard errors estimator, and both unstandardized (with its relative confidence intervals) and standardized coefficients were reported. The parameters of SEM were estimated by bootstrapping over 1000 samples to make the model fitting robust to non-normal data. Bias-corrected bootstrap confidence intervals were implied to test indirect and total effects. Bootstrap confidence intervals were reported with each estimated coefficient and its related significance test. As the model was fully saturated (i.e., they perfectly fitted the data because they had as many parameters as there were values to be fitted), no goodness-of-fit scores were reported. This analysis should be considered exploratory as it aimed to show how mixing implicit and explicit measures could lead to new insights about the relationship with the construct of interest.

## 3. Results

In this section, we provided data analysis to support the usefulness of the V-IRAP. We firstly demonstrated that the task was reliable (Section 3.1), and then we showed how implicit attitudes biased responses towards the positive or negative evaluation of the vaccination (Section 3.2). We also compared such biases in the vaccinated and non-vaccinated participants (Section 3.3). In the next sections, the relationship between the implicit measures obtained with the V-IRAP and the explicit measures of vaccine acceptance was investigated through correlation (Section 3.4) and regression analysis (Section 3.5). The last section (Section 3.5) reported an exploratory mediation analysis showing how implicit biases influenced vaccination behavior through their effects on explicit vaccine acceptance. Overall, we showed how implicit measures went beyond explicit ones, as supported in particular by both the incremental validity analysis and the mediation model.

### 3.1. Preliminary Analysis

In terms of experimental attrition, of the total of 151 participants, one did not reach the accuracy standard of at least 70% on all test blocks. We also removed 13 participants with median RT slower than 2500 ms, and 3 more for responding with RT < 300 ms in at least 10% of trials. Then, we removed 17 participants in total from the starting sample. The exact Fisher test was conducted to examine differential attrition with respect to vaccination behavior. The test was not significant, *p* = 0.53, indicating similar attrition for vaccinated (12 removed) and non-vaccinated (5 removed) participants. We obtained a final sample for the analysis with 134 participants (females = 110, males = 24; mean age = 37.26 years, SD = 12.45; education = 18.20 years, SD = 2.92). A total of 105 participants had received a COVID-19 vaccine, while 28 had not. Overall, in this sample, the average accuracy was 92.41% and the median RT was 1782.45 ms.

To assess the internal reliability of the IRAP task, we computed the Pearson correlation between the overall D-score obtained in even and odd trials. The correlation between the scores was significant, r = 0.32, *p* < 0.01, with a Spearman–Brown corrected coefficient resulting in r = 0.49.

### 3.2. IRAP Effects

The D-scores for the four trial types were calculated, that is, COVID-19-positive, COVID-19-negative, VACCINE-positive, and VACCINE-negative. As shown in Figure 2, the average D-scores were positive, indicating a bias toward the consistent rule, that is, COVID-19 was negative and VACCINE was positive. Then, a series of one-sample *t*-tests on the four IRAP D-scores separately was conducted to assess the significance and the strength of each bias. For each analysis, we report the average score, 95% confidence intervals, *t*-test statistics, and Cohen’s effect size as d. The analysis revealed a significant positive bias for the COVID-19-negative trial type D-score, M = 0.22 [0.15, 0.29], t(143) = 6.02, *p* < 0.01, d = 0.50. It also revealed significant positive biases for both VACCINE-negative, M = 0.24 [0.17, 0.31], t(143) = 6.48, *p* < 0.01, d = 0.54, and VACCINE-positive D-score, M = 0.36 [0.39, 0.42], t(143) = 11.18, *p* < 0.01, d = 0.93. The D-score relative to the COVID-19-positive trial type was not significantly biased, M = 0.05 [−0.02, 0.11], t(143) = 1.48, *p* = 0.14, d = 0.12. Thus, all significant biases were in the expected direction and with a moderate to large effect size.

The D-scores obtained in the four trial types were then compared through a one-way ANCOVA while controlling for participants’ age, sex, and education level in this analysis. The results (Figure 2) revealed a significant main effect of trial type, F(3, 565) = 13.87, *p* < 0.01. Holm–Bonferroni corrected post hoc *t*-tests revealed that bias for COVID-positive trials was significantly lower than all the others, *p* < 0.01 for all comparisons, while the VACCINE-positive bias was greater than all others, *p* < 0.01 for all comparisons. The biases for VACCINE-negative and COVID-19-negative trial types were similar, *p* = 0.64. 

As we were mainly interested in the implicit bias towards vaccines, for the subsequent analyses, only the D-scores relative to the vaccination label were considered—that is, the VACCINE-positive and VACCINE-negative D-scores.

### 3.3. Comparisons for Vaccination Behavior

The biases obtained for participants who were vaccinated against COVID-19 (N = 105) and participants who were not (N = 28) were compared. Figure 3 shows the average D-scores obtained in the two groups. A 2 (VACCINE-positive vs. VACCINE-negative) × 2 (vaccinated vs. non-vaccinated) mixed ANCOVA was conducted on D-scores, controlling for age, sex, and education level of participants. The ANCOVA revealed a significant main effect of trial type, F(1, 279) = 6.26, *p* < 0.05, as for the main analysis, and a significant main effect of the vaccination group, F(1, 279) = 8.59, *p* < 0.01, with lower overall D-scores for non-vaccinated participants (M = 0.22) than for vaccinated ones (M = 0.40). The interaction effect between trial type and vaccination group did not reach significance, *p* = 0.88. As also reported in Figure 3, the main pattern of results was comparable in the two groups, but the non-vaccinated participants reported D-scores closer to zero for all the trial types. 

Holm–Bonferroni corrected two-sample *t*-tests confirmed this result. Comparing the average D-scores between trial types revealed a significant difference between the groups for the VACCINE-positive D-scores, M_1_ = 0.39, M_2_ = 0.22, t(141) = 2.34, *p* < 0.05, and an almost significant difference for the VACCINE-negative D-score, M_1_ = 0.28, M_2_ = 0.11, t(141) = 1.83, *p* = 0.07. Thus, non-vaccinated participants reported lower overall D-scores than vaccinated ones.

To deepen this analysis, the strength of response biases was also assessed for the two groups separately, with a series of one-sample *t*-tests on the trial types D-score for each group. Regarding the vaccinated group, the analysis revealed significant positive biases for both VACCINE-negative, M = 0.27 [0.19, 0.36], t(111) = 6.74, *p* < 0.01, d = 0.64, and VACCINE-positive D-scores, M = 0.40 [0.33, 0.47], t(111) = 11.46, *p* < 0.01, d = 1.09, thus with moderate and large biases, respectively, for the relationships vaccine as not harmful and vaccine as positive/healthy. Regarding the non-vaccinated participants, the analysis showed a positive bias for the VACCINE-positive D-score, M = 0.22 [0.07, 0.37], t(32) = 3.00, *p* < 0.01, d = 0.53, but no bias for the VACCINE-negative trial type, M = 0.11 [0.05, 0.28], t(31) = 1.40, *p* = 0.17, d = 0.25. Then, they tended to respond equally fast in the VACCINE-negative trials with both rule A and rule B, and they also showed a weaker bias than vaccinated participants in responding accordingly to rule A for the VACCINE-positive trial.

### 3.4. Correlation Analysis and Relationship with Explicit Measures

Here, we present the results of the Pearson correlation analysis conducted between the D-scores obtained in the IRAP with the explicit measures of vaccine acceptance. First, a correlation analysis between the two D-scores obtained for the VACCINE-positive and VACCINE-negative trials revealed a positive but weak correlation, r = 0.20, *p* < 0.05, indicating only a partial overlap between the positive and negative aspects of bias toward vaccines. Regarding the correlation with the explicit evaluation of vaccine acceptance, the analysis revealed a positive correlation with the VACCINE-negative D-score, r = 0.23, *p* < 0.01, but no correlation with the VACCINE-positive D-score, r = 0.04, *p* = 0.64. Again, this pattern of results confirmed a differential role for positive and negative biases towards vaccination. 

### 3.5. Incremental Validity with Hierarchical Regression Analysis

In this section, we present the results relative to the incremental validity analysis in which we assessed the relative contribution of the explicit and implicit vaccine acceptance as measured using the questionnaire and V-IRAP task, respectively. To achieve this aim, a hierarchical regression model was applied to assess their predictivity effects on vaccination behavior (0 = not vaccinated; 1 = vaccinated). The results of this analysis are reported in Table 1.

At step 1, only education level was a significant predictor of vaccination, b = 0.06, *p* < 0.01. At step 2, vaccine propensity was a significant predictor of vaccination, b = 0.10, *p* < 0.01, with a significant increase in model fit, ΔR^2^ = 0.08, *p* < 0.01. In step 3, adding the implicit measures, we found that only the D-score of the VACCINE-positive trials was a significant predictor of vaccination, b = 0.21, *p* < 0.01, while the D-score of the VACCINE-negative trials was not, b = 0.05, *p* = 0.48. In general, the addition of implicit measures significantly increased the model fit, ΔR^2^ = 0.04, *p* < 0.05. This analysis showed how implicit bias as measured in the IRAP task was a significant predictive factor of actual vaccination behavior beyond explicit measures of vaccine propensity.

### 3.6. Exploratory Mediation Analysis

As the last analysis, we proposed a mediation model. Based on previously reported results, we hypothesized that implicit bias towards vaccination influenced explicit evaluation of vaccines and that this mediated the effect of implicit biases on actual vaccination. In particular, implicit bias for positive aspects of the vaccine seems to directly increase vaccination adherence, whereas fear of vaccination in terms of implicit bias for negative aspects of the vaccine seems to reduce vaccination behavior through a mediation effect of the explicit evaluation regarding vaccines. These hypothesized paths were based (1) on the positive correlation between the D-score for VACCINE-negative trials and explicit vaccine propensity, (2) on the absence of correlation between the D-score for VACCINE-positive trials and explicit vaccine propensity, and (3) on the regression results, in which the D-score for VACCINE-negative showed no effect on vaccine behavior when controlling for explicit vaccine propensity, while the D-score for VACCINE-positive trials showed a positive predictive effect. 

Table 2 reported the estimated coefficients of the structural equation model, and Figure 4 depicted the structural paths with their standardized coefficients. As shown, the D-score for VACCINE-negative trials significantly increased explicit propensity to the vaccine, and this, in turn, increased vaccination behavior. Instead, the D-score for VACCINE-positive trials did not correlate significantly with the mediator and directly and positively affected vaccination behavior. Thus, the effect of implicit bias for negative aspects on vaccination seemed to be mediated by the explicit propensity to vaccination, while implicit bias for positive aspects showed a direct effect. Test of indirect effects through bias-corrected bootstrapped confidence intervals confirmed this dissociation: the mediated path from D-score for VACCINE-negative trials to vaccination through explicit propensity was significant, while no significant mediation existed for VACCINE-positive D-scores.

## 4. Discussion

This study aimed to investigate the relationship between implicit and explicit attitudes towards vaccination. We developed and validated a new implicit test, V-IRAP, evaluating the strength of relational links of positive and negative target words with COVID-19 vaccination. To our knowledge, this is the first study to use an IRAP task in the area of vaccine hesitancy attitudes. The V-IRAP task presented in this paper showed good internal consistency and reliability, as well as concurrent and discriminant validity. Notably, we tested the V-IRAP on a sample larger than those in previous IRAP experiments, which usually involved on average about 50–70 participants [41,42]. Implicit attitudes measured on V-IRAP significantly correlated with explicit evaluation of vaccine hesitancy but also showed an incremental validity in predicting actual adherence to the vaccination campaign. This result suggests that both explicit and implicit evaluations on vaccine hesitancy measured the same construct, but probably at different levels of awareness. The first could be verbalized while the second tapped into a deeper level, revealing inner attitudes of which individuals may not be completely aware. Although the IRAP procedure is a relatively direct or explicit measure of cognition, performance could not be faked even when participants were instructed to do so [43]. Therefore, the outcome of V-IRAP in terms of positive and negative attitudes to vaccination can be considered a reliable and compelling measure of implicit beliefs about vaccination. In addition, the V-IRAP reveals the deep attitudes of the participants toward vaccination and how these attitudes contribute to hesitancy or acceptance of the vaccine. The technique of implicit associations has made it possible to detect how vaccination involves complex psychological factors that lead to a simple declaration of accepting or to rejecting the vaccine.

Overall, participants reported a bias toward considering the vaccination as positive in V-IRAP. This result was in line with the recent literature on the acceptance rate of the COVID-19 vaccine, which is very high in Italy. Furthermore, this result supports the view that people mainly accept the vaccine because they feel it is safe and useful in fighting the spread of the disease [44]. Another possible explanation was that people actually fear the infection and thus act to reduce the risk of being infected or contracting the disease [26]. This alternative or concurrent explanation appeared to be supported by the positive bias shown for the relational COVID-negative, indicating that people tended to associate the disease with negative targets more easily than with positive ones. In light of this result, the study supports previous results obtained with self-report measures in which individuals showed higher compliance to vaccination as their risk perception towards the disease increased [23,45], or if they felt the vaccine was safe or not harmful [27]. Thus, our study showed that the population tends to get vaccinated because they trust the vaccination and fear the disease or the infection [2].

The task did not highlight other compelling explanations or drives for vaccine acceptance. First, social factors were not investigated in this study. Sympathy for people who could not be vaccinated for health-related problems would be a positive drive for vaccine acceptance [46]. Interestingly, this aspect was also strongly underlined in the COVID-19 vaccination communication campaign, in which the request to take the vaccine as a moral or community duty was stressed. However, prosocial and emphatic attitudes could not be changed so easily, thus suggesting it is better to rely both on socially and personally oriented communication campaigns [47]. According to a study by Freedman et al. [48] conducted on the UK population, information on personal benefits reduces hesitancy to a greater extent than information on collective benefits. As such, a message that stresses the personal benefits is likely to be most effective, while messages emphasizing the risks of the virus, or the safety of vaccination, had no effect on vaccination intentions [49].

On this point, a second factor that this study did not consider was the mistrust in science and the role of conspiracy theories, which are both important factors in determining vaccine hesitancy [2,10,50]. To this end, future studies that involve the V-IRAP could assess the relationship between implicit attitudes towards vaccines and these constructs, measured with explicit or implicit methods. In this regard, a possible future direction of this study could be to use a similar IRAP procedure to test the relational cognition of conspiracy theories, addressing the bias towards considering such theories as true or false.

The comparison between participants who get vaccinated and participants who do not reveals an interesting pattern of results. The V-IRAP showed that the main difference between these two groups of participants was based on the evaluation of the harmful aspects of vaccines. Participants who did not get vaccinated showed no bias for the association between the label ‘VACCINE’ and negative targets, while vaccinated participants reported a bias towards a false response for such an association. Furthermore, this VACCINE-negative association also correlated with explicit evaluation of the vaccines, while the VACCINE-positive association did not. Taken together, these results suggest that hesitant people were primarily concerned about the possible damages derived from vaccination and that these concerns concurred in forming their explicit evaluation of the vaccine. This result is in line with the previous literature, which reported the central role of risk perception about vaccines as a predictor of acceptance or hesitancy [8,18]. Fear of having serious or irreversible consequences after vaccination may be associated with the fear of death, increasing hesitancy. These results were consistent with other studies in which the main barriers to the COVD-19 vaccine were the perception of limited efficacy, potential adverse effects, and safety concerns [51,52,53]. Therefore, the present data suggested that a future information campaign should focus more on reducing the fear of side effects of vaccines than on enhancing their capacity to prevent infection. This strategy was already reported as successful in increasing pediatric vaccination [28], and should be considered also for future pandemic-related vaccination campaigns. 

However, non-vaccinated participants also reported a reduced bias toward the VACCINE-positive association—that is, they considered the vaccine as less useful than participants who accepted the vaccination. Interestingly, positive aspects related to vaccination were a positive and direct predictor in both the regression model and the mediation model. That is, underlining the positive aspects of vaccination would increase vaccine acceptance also among hesitant individuals. Mediation analysis showed that considering the vaccine to be safe and healthy directly increases vaccination behavior according to the literature [54]. Thus, beliefs about the usefulness of vaccination could increase the probability to get vaccinated [55]. 

The lack of mediated effect through explicit vaccine evaluation suggested that in general, participants considered vaccines useful in fighting the pandemic, but that negatives and aspects of risk connected to vaccination could oppose acceptance by forming an explicit, critical position toward vaccination. Motivational aspects related to avoiding vaccine-related risks seemed to prevail over motivation to avoid disease-related risks. Again, individuals showed difficulty in evaluating the correct risk–benefit ratio associated with vaccination [56] and appeared to rely more on emotional factors to drive their decision to acceptance or refusal of vaccination [2]. Therefore, an effective vaccination campaign should emphasize emotional aspects related to vaccination more than cognitive ones [30]. Future studies on this topic could directly test this hypothesis by measuring implicit and explicit vaccination attitudes after different inductions: in one experimental condition, participants would be given or not given information about the effectiveness of vaccination and its advantage over getting the infection, while in another condition, participants would receive or not receive an emotional induction on vaccine-related risk. The present results support the hypothesis that manipulation at a cognitive level would affect explicit evaluation, while implicit attitudes only seem to be affected by emotional manipulation.

The results of this study should be interpreted in the light of some limitations. First, the experimental procedure was conducted completely online. While the reliability of online experiments has been extensively proven [57], the IRAP is a complex task that involves some performance parameters to respect, such as being fast and accurate enough. In this respect, the presence of the experimenter in the practice sessions could ensure that participants correctly understand the instructions. However, it should be noted that in this experiment, attrition was not consistent, but we had also relaxed the usual IRAP performance constraints. This is known to reduce internal reliability [41]. Thus, future replications of the same procedure in a laboratory-controlled environment with more severe constraints would be welcomed. A second limitation can be identified in the convenience sampling methods involved in the data collection, which did not allow us to control many variables. For example, the present sample was unbalanced with regard to the demographic factor of sex, with a greater presence of women than males. As female sex was considered a predictor of vaccine hesitancy [58], an unbalanced sample may alter the pattern of effects revealed and so undermine the generalizability of the results. In the present study, in order to minimize the possible influence of such a limitation, all analyses were corrected for demographic variables. However, a more refined sampling method should be used in future research on this topic. 

Notwithstanding these possible limitations, the current findings suggest that the V-IRAP is a sensitive and reliable tool that could be used in future studies on implicit attitudes towards vaccination. In particular, the same procedure could also be conveniently applied to study the acceptance of other vaccinations, such as the annual influenza vaccination in adults or pediatric vaccinations. The label ‘COVID-19’ could be substituted with another disease to be vaccinated against, for example, ‘MEASLES’ or ‘FLU’. The tool would again reveal which implicit beliefs would push individuals towards acceptance or refusal of vaccination.

## 5. Conclusions

The study of psychological causes of vaccine hesitancy is one of the most important aims of modern psychological research, with a great impact on social life and global health. Refining the tools we can use as scientists to investigate this construct is a step forward to a greater understanding of such a relationship. The V-IRAP is a new instrument that allows the collection of data about implicit biases towards vaccinations, and is capable of distinguishing between positive and negative beliefs related to vaccination. Understanding the deep causes of vaccine hesitancy could increase the possibility of arranging successful vaccination campaigns that are centered on the needs of people and their emotional lives [8]. The V-IRAP could become in the future a standardized method for the assessment of vaccine hesitancy [15]. The results reported here have many practical implications. First, they show that hesitant people tend to have a negative evaluation of the vaccines which could be linked to a distrust in medical science [2]. Increasing people’s trust in medical science may be an important avenue of investment for increasing adherence to future vaccination campaigns. Second, the implicit bias towards negative aspects contributes to the explicit evaluation of vaccination, while the implicit positive bias does not. Hence, these negative aspects would more probably be spread in the anti-vaccination groups or echo chambers on the Internet. Therefore, an effective communication campaign should be informed by this outcome in order to prepare appropriate and targeted messages. As shown, the V-IRAP allows the investigation of the unspoken reasons behind vaccine hesitancy, in order to prepare better remediations and plan more effective communication campaigns.

## Figures and Tables

**Figure 1 ijerph-19-04205-f001:**
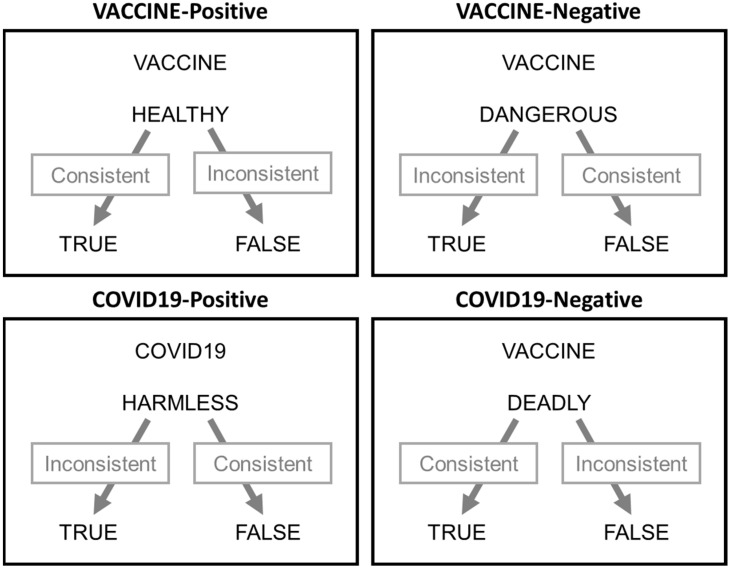
The four trial types as a combination of labels (VACCINE and COVID-19) and targets (positive or negative adjectives). The response options are shown at the bottom. Grey boxes and arrows were not presented and indicate which responses are correct in consistent or inconsistent blocks.

**Figure 2 ijerph-19-04205-f002:**
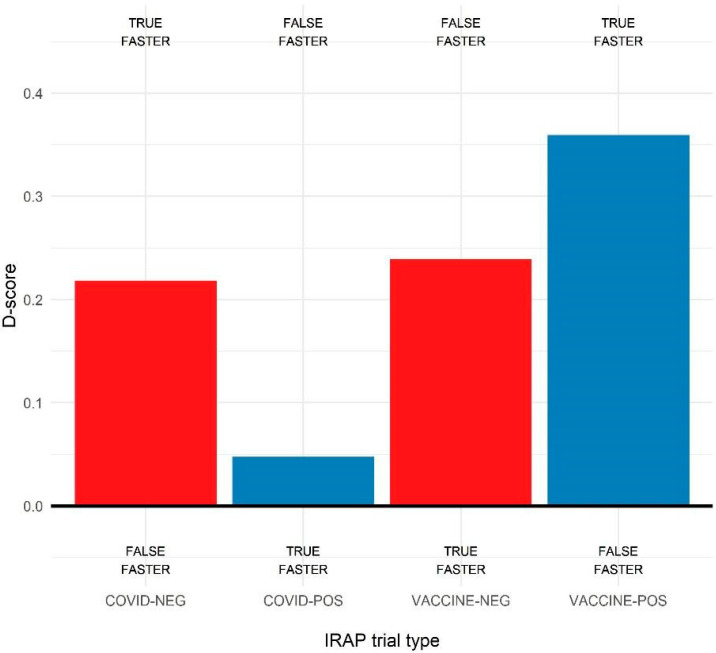
Average D-scores obtained in the four IRAP trial types, with the combination of labels (VACCINE and COVID-19) and targets (POSitive or NEGative). Labels at the top or bottom of the plot indicate the direction of the bias according to the consistent (upper) or inconsistent (lower) rules. As shown, all D-scores were positive, i.e., participants responded faster in the consistent block for all the four trial types.

**Figure 3 ijerph-19-04205-f003:**
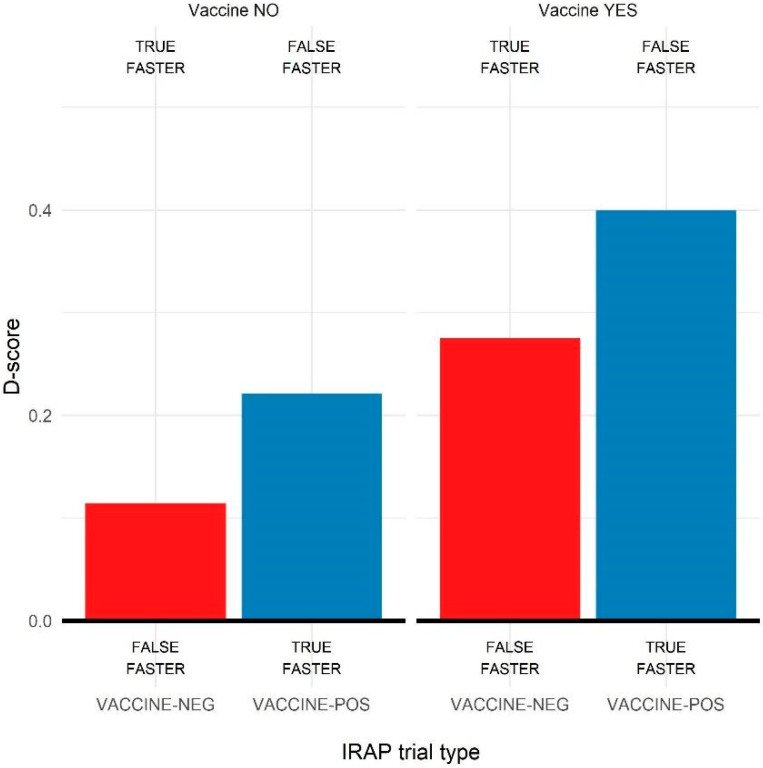
Average D-scores obtained in the four IRAP trial types, with the combination of labels (VACCINE and COVID-19) and targets (POSitive or NEGative), by non-vaccinated (**left** panel) and vaccinated (**right** panel) participants. The labels at the top or bottom of the plot indicated the direction of the bias according to the consistent (upper) or inconsistent (lower) rules.

**Figure 4 ijerph-19-04205-f004:**
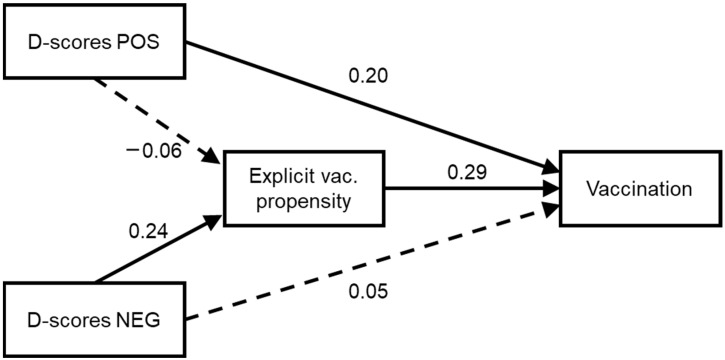
The mediation model tested. Continuous arrows represent significant paths, while dotted arrows represent non-significant paths. Indirect effects are reported in the text. The effects of covariates (sex, age, and education level) are not reported for the sake of clarity. D-score POS = D-scores for VACCINE-positive trials; D-score NEG = D-scores for VACCINE-negative trials.

**Table 1 ijerph-19-04205-t001:** Regression results using vaccine behavior as a dependent variable.

Model	Predictor	b	b 95% CI	β	β 95% CI	Fit	Difference
Step 1	(Intercept)	−0.37	[−0.85, 0.10]				
	Sex	0.12	[−0.08, 0.31]	0.11	[−0.07, 0.29]		
	Age	−0.00	[−0.01, 0.00]	−0.02	[−0.22, 0.16]		
	Education	0.06 **	[0.04, 0.08]	0.42	[0.25, 0.58]		
						R^2^ = 0.19 **	
						95% CI [0.10, 0.36]	
Step 2	(Intercept)	−0.83 **	[−1.27, −0.36]				
	Vaccine propensity	0.10 **	[0.04, 0.15]	0.30	[0.11, 0.46]		
						R^2^ = 0.28 **	ΔR^2^ = 0.08 **
						95% CI [0.17, 0.44]	95% CI [0.02, 0.19]
Step 3	(Intercept)	−0.94 **	[−1.37, −0.45]				
	D-score POS	0.21 **	[0.06, 0.36]	0.20	[0.06, 0.33]		
	D-score NEG	0.05	[−0.08, 0.18]	0.05	[−0.08, 0.18]		
						R^2^ = 0.32 **	ΔR^2^ = 0.04 *
						95% CI [0.22, 0.48]	95% CI [0.01, 0.11]

Note. A significant b-weight indicates that the beta-weight is also significant. b represents unstandardized regression weights. β indicates the standardized regression weights. Bootstrapped lower and upper limits of a confidence interval are reported in square brackets. D-score POS = D-score for VACCINE-positive trials; D-score NEG = D-scores for VACCINE-negative trials. The significant level is indicated as follows: * *p* < 0.05, ** *p* < 0.01.

**Table 2 ijerph-19-04205-t002:** Structural path coefficients, indirect and total effect for the mediation model.

Path			b	CI Lower	CI Upper	SE	β
D-score NEG	→	Vaccination	0.05	−0.06	0.19	0.07	0.05
D-score POS	→	Vaccination	0.21 **	0.06	0.35	0.07	0.20
D-score NEG	→	Explicit prop.	0.72 **	0.21	1.21	0.26	0.24
D-score POS	→	Explicit prop.	−0.21	−0.80	0.44	0.32	−0.06
Explicit prop.	→	Vaccination	0.09 **	0.05	0.15	0.03	0.29
Indirect effect of D-score POS			0.07 *	0.02	0.15	0.03	0.07
Indirect effect of D-score NEG			−0.02	−0.09	0.03	0.03	−0.02
Total effect			0.31 **	0.09	0.50	0.11	0.30

Note. b = unstandardized coefficient, CI lower and CI upper = lower and upper 95% bootstrapped confidence intervals of b, SE = standard error, β = standardized coefficient. D-score POS = D-scores for VACCINE-positive trials; D-score NEG = D-scores for VACCINE-negative trials. The significant level is indicated as follows: * *p* < 0.05, ** *p* < 0.01.

## Data Availability

The data that support the findings of this study are available on request from the corresponding author.

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
