# Peer review of "How Implicit Attitudes toward Vaccination Affect Vaccine Hesitancy and Behaviour: Developing and Validating the V-IRAP"

_ijerph, 2022, doi:10.3390/ijerph19074205_

Round 1

Reviewer 1 Report

Dear Authors,

The theme of the work carried out is pertinent and falls within the scope of the International Journal of Environmental Research and Public Health.

The study is very important and explores a current, relevant and emerging issue in society.

As a reviewer, I suggest the following comments to maximize the strength of the article:

Introduction:

  1. The manuscript did not consider the large amount of already existing publications on vaccination against COVID-19. Authors should carefully revise this existing literature. I encourage authors in explicating the relationships with previous literature and draw from it a number of significant and interesting hypotheses that could justify the interest in reading this paper.
  2. You cite the studies on the acceptance of vaccination in the US, China and Australia, while in other geographic areas there are more unvaccinated people (lines 80-82). Please write how the researchers explain this fact.

Materials and Methods:

  1. Please write down when the research was performed.
  2. Please verify the information "Regarding the vaccination for COVID-19, 118 (78%) participants took at least a dose of the vaccine, while 33 (32%) did not" (lines 163-164). 33 unvaccinated people do not constitute 32% of the respondents.

Results:

The following stages of statistical analyzes are described in detail, but after reading this part of the manuscript, I do not know what I actually learned.

It may be worthwhile to make a summary that will allow the reader to get an overall picture of what has been found.

Conclusion:

In the conclusion, authors should describe the main practical implications of their study.

Best regards

Author Response

We wish to thank the Reviewer for the appreciation of our work and the useful suggestions. Please find enclosed the point-to-point answer (in red) to the raised issues.

Introduction:

  1. The manuscript did not consider the large amount of already existing publications on vaccination against COVID-19. Authors should carefully revise this existing literature. I encourage authors in explicating the relationships with previous literature and draw from it a number of significant and interesting hypotheses that could justify the interest in reading this paper.
    We are aware that the COVID-19 literature on vaccination hesitancy is more extended than what we reported in the manuscript. While it is not possible nor useful to produce an exhaustive review of the existing literature in our introduction, we added some relevant systematic reviews on this topic to summarize the extent of the scientific knowledge in the field. We also added a number of explicit hypotheses at the end of the introduction as follows “From the reviewed literature, we hypothesized in particular that non-vaccinated participants should report both a belief that vaccination was not safe [18], that is they would report a bias towards the association between vaccine and negative aspects, and a belief that vaccination is not useful [14], that is they would report a reduced bias towards the association between vaccine and positive aspects. Moreover, we hypothesized that implicit attitudes were predictive of vaccination hesitancy and that they also increment the capability of predicting vaccination behavior with respect to the mere explicit measure.”

    We added the following references:

    - Sallam, M. COVID-19 Vaccine Hesitancy Worldwide: A Concise Systematic Review of Vaccine Acceptance Rates. Vaccines 2021, 9, 160. https://doi.org/10.3390/vaccines9020160

    - Aw, J.; Seng, J.J.B.; Seah, S.S.Y.; Low, L.L. COVID-19 Vaccine Hesitancy—A Scoping Review of Literature in High-Income Countries. Vaccines 2021, 9, 900. https://doi.org/10.3390/vaccines9080900

    - Biswas, M.R.; Alzubaidi, M.S.; Shah, U.; Abd-Alrazaq, A.A.; Shah, Z. A Scoping Review to Find Out Worldwide COVID-19 Vaccine Hesitancy and Its Underlying Determinants. Vaccines 2021, 9, 1243. https://doi.org/10.3390/vaccines9111243

    - Troiano G, Nardi A. Vaccine hesitancy in the era of COVID-19. Public Health. 2021;194:245

    - : Patwary, M.M.; Alam, M.A.; Bardhan, M.; Disha, A.S.; Haque, M.Z.; Billah, S.M.; Kabir, M.P.; Browning, M.H.E.M.; Rahman, M.M.; Parsa, A.D.; et al. COVID-19 Vaccine Acceptance among Low- and Lower-Middle-Income Countries: A Rapid Systematic Review and Meta-Analysis. Vaccines 2022, 10, 427. https://doi.org/10.3390/ vaccines10030427

    - Zintel, S., Flock, C., Arbogast, A.L. et al. Gender differences in the intention to get vaccinated against COVID-19: a systematic review and meta-analysis. J Public Health (Berl.) 2022. https://doi.org/10.1007/s10389-021-01677-w

    - Norhayati MN, Che Yusof R, Azman YM. Systematic Review and Meta-Analysis of COVID-19 Vaccination Acceptance. Front Med (Lausanne). 2022;8:783982. Published 2022 Jan 27.doi:10.3389/fmed.2021.783982

    - Zeng, Z.; Ding, Y.; Zhang, Y.; Guo, Y. Title. Int. J. Environ. Res.Public Health in press. doi:10.1016/j.puhe.2021.02.025

  2. You cite the studies on the acceptance of vaccination in the US, China and Australia, while in other geographic areas there are more unvaccinated people (lines 80-82). Please write how the researchers explain this fact.
    Our research did not have a direct explanation for such fact. Following this reflection, we removed this sentence while leaving in the manuscript the statistics relative to the Italian situation, so to give the readers the context in which the research was conducted.

 Materials and Methods:

  1. Please write down when the research was performed.
    We added this information in the revised version of the manuscript, Section 2.1: “For this study, we enrolled 151 volunteer participants through snowball sampling on social media during May, June, and July 2021.”
  2. Please verify the information "Regarding the vaccination for COVID-19, 118 (78%) participants took at least a dose of the vaccine, while 33 (32%) did not" (lines 163-164). 33 unvaccinated people do not constitute 32% of the respondents.
    Thanks for finding this error. The correct percentage (22%) is reported in the revised manuscript.

Results:

The following stages of statistical analyzes are described in detail, but after reading this part of the manuscript, I do not know what I actually learned. It may be worthwhile to make a summary that will allow the reader to get an overall picture of what has been found.
Thanks for this useful suggestion. Following this, we added a short new paragraph at the beginning of the Results section to give a summary of the reported analyses, as follows: “In this section, we provided data analysis to support the usefulness of the V-IRAP. We firstly demonstrated that the task was reliable (Section 3.1), then we showed how implicit attitudes biased responses towards the positive or negative evaluation of the vaccination (Section 3.2). We also compared such biases in the vaccinated and non-vaccinated participants (Section 3.3). In the next sections, the relationship between the implicit measures obtained with the V-IRAP and the explicit measures of vaccine acceptance was investigated through correlation (Section 3.4) and regression analysis (Section 3.5). The last section (Section 3.5) reported an exploratory mediation analysis showing how implicit biases influenced vaccination behavior through their effects on explicit vaccine acceptance. Overall, we showed how implicit measures went beyond explicit ones, as supported in particular by both the incremental validity analysis and the mediation model.”

Conclusion:

In the conclusion, authors should describe the main practical implications of their study.
Following this suggestion, we added some practical implications of our study in the Conclusions section as follows: “The results reported here have many practical implications. First, it showed that hesitant people tend to have a negative evaluation of the vaccines which could be linked to distrust in medical science [2]. Increasing people's trust in medical science may be an important avenue of investment for increasing adherence to future vaccination campaigns. Second, the implicit bias towards negative aspects contributes to the explicit evaluation of vaccination, while the implicit positive bias does not. Hence, these negative aspects would be more probably spread in the anti-vaccination groups or echo chambers on the internet. Therefore, an effective communication campaign should be informed by this outcome in order to prepare appropriate and targeted messages. As shown, the V-IRAP allows investigation of the unspoken reasons behind vaccine hesitancy, in order to prepare better remediations and plan more effective communication campaigns.”

Reviewer 2 Report

I found the article well structured and well written, which facilitated the reading. The introduction was motivating and clear and the discussion was written in a way that allowed to interpret the scope of the results. The topic is really interesting and the approach novel. Besides minor clarifications (attached) I think the article is in good shape for publication.

Author Response

We wish to thank the Reviewer for the appreciation of our work. We took into consideration all your comments in the revised version of the manuscript.

First, we added a clarifying note on participants who agreed to complete both questionnaires and IRAP task in the procedure “As reported, 151 participants completed both the questionnaires and the IRAP task, while 224 completed only the questionnaires (their data were not reported here).”

Second, we clarify what constituted a correct or wrong response in the IRAP task by adding the following sentences to Section 2.3.2: “A correct response, i.e., a response in accordance with the association rule given at the beginning of the block, cleared the screen for 400 ms and then another trial was presented. In the case of a wrong response, i.e., a response not in accordance with the association rule given at the beginning of the  block, a red X was presented between the two response options until the participant gave the correct response.

About targets (as commented in Figure 1), we are aware that better words maybe existed. However, we conducted a careful decision process for selecting those words. It is possible that the Italian-English translation slightly alters the original meaning of the selected words. Thus, we also added to the manuscript the list of Italian words implied in the task: “… the targets were divided into two 5-item lists, one including adjectives indicating a positive attitude or trust, e.g., good, healthy, useful, harmless, and reassuring (in the original Italian version: buono, salutare, utile, innocuo, rassicurante), and the other including negatively connotated adjectives, e.g. bad, deadly, harmful, dangerous, and frightening (in the original Italian version: cattivo, mortale, dannoso, pericoloso, spaventoso).”
We are also aware that no one would respond that COVID was good or healthy, but please consider that a) the task aims is to tap into inconsistent or anti-intuitive associations, and b) that we are marginally interested in the COVID-related response. In fact, we focused on the VACCINE-related responses. We better clarify this aspect in the revised version by adding the following sentence at the end of Section 2.3.2: “While the association between COVID-19 and terms such as good or healthy could be considered bizarre, please consider that the task aimed properly to reveal anti-intuitive or unusual associations. Moreover, as we were mainly interested in the responses associated with the labels 'VACCINE' (see Results section), such associations with COVID-19 were not problematic to us.”